# Postharvest Geometric Characterization of Table Olive Bruising from 3D Digitalization

Ramón González-Merino [1] , Rafael E. Hidalgo-Fernández [2] , Jesús Rodero [1] , Rafael R. Sola-Guirado [3] and Elena Sánchez-López [2,*]

1   Department of Visual Computing, Technology Centre of Metal-Mechanical and Transport, 23700 Linares, Spain
2   Department of Graphic and Geomatic Engineering, Campus of Rabanales, University of Córdoba, 14071 Cordoba, Spain
3   Department of Mechanics, Campus de Rabanales, University of Cordoba, 14071 Cordoba, Spain
*   Correspondence: elena.sanchez@uco.es

**Abstract:** The physical properties of table olive fruit are an important factor in the design of harvesting, transport, classification, and commercialization. The visual quality of the fruits harvested is the most important factor limiting the commercialization of table olives. The mechanical damage during harvesting consists of local tissue degradation, resulting in bruising of the fruits. In recent years, several studies have been carried out to identify physical properties and to calculate indices that characterize the damage to olives. However, all of them are based on 2D techniques. The aim of this work is the determination of new geometric parameters based on a 3D analysis of the scanned olives. The 3D shape parameters have been collated with those obtained by standard 2D shape analysis methods. From the results, it is observed that the use of high-resolution, medium-cost 3D technologies allows a more precise characterization of the shape of damages observed in table olives. To carry out three-dimensional analysis, Boolean operations of the solid and parametric surfaces of the meshes obtained by a 3D scanner have been used.

**Keywords:** 3D scanning; table olives; fruit damage; dimensional properties; modelling

## 1. Introduction

Spain is the world's leading producer of table olives (*Olea europaea* L.), a market that every year generates worldwide trade valued at EUR 1700 million [1]. The world production of table olives is approximately 2,504,000 t. The cultivation of table olives presents a great diversity of productive situations, so in the analysis of production costs, they must be established according to the different types of cultivation and the costs derived from each of them. Of the main costs, the vast majority are common to olives intended for oil production and table olives, such as the costs of fertilization, soil maintenance, pruning, and phytosanitary treatments, among others. The main difference is found in the costs derived from the harvesting technique. In general, this first stage of the production process entails an increase in costs due to the fact that the highest percentage of table olive harvesting is carried out by hand picking methods. Other harvesting alternatives use semi-mechanized systems, such as branch shakers and shaker combs [2], but they contribute to the further development of damage to the olive.

The incorporation of mechanization techniques for harvesting is strongly hindered due to the different damages suffered by the olive throughout the process. In general, the olives receive blows during the harvesting process, handling operations, and post-harvest transport, giving rise to a phenomenon called "molestado" [3]. Fruit bruising consists of a cellular tissue rupture that releases intracellular water, leading to the oxidation of phenolic compounds. In time, depending on the characteristics of the impact, the affected skin darkens and contrasts notably with the rest of the olive's green color [4]. In the

case of olives dedicated to oil extraction, this phenomenon is not important, but for table olives it represents a very important handicap in terms of product quality, so much so that the minimum quality criteria and the defects that this type of olives may have for their commercialization are determined by the Commercial Standard COI/OT/NC nº1 of December 2004 and Royal Decree 679/2016 and Royal Decree 679/2016 [5,6]. In the current marketing system, the quality and appearance of the fruit prevail over other parameters, such as the proportion of the pulp versus the pit, or the non-use of phytosanitary products.

In general, there are a number of factors that favor the appearance of damage to the fruit. The incidence and severity of damage is related to various pre-harvest and post-harvest causes [7–11]. Several studies have focused on determining factors intrinsic to the fruit, such as shape, size, amount of water, firmness, intercellular strength, elasticity, shape, and cell structure, among others, as possible internal factors [9,12], and these are related to the appearance of these damages.

The study and characterization of these factors has been the main objective of many of the research works on this subject that have been developed in recent decades. Many authors have experimented with various techniques to determine the origin of this damage, as well as the power to establish some type of index that allows measuring and characterizing the damage with the aim of preventing it. In summary, in recent decades various types of tests have been carried out to determine the internal properties and detect damage to the fruit. Among this type of test are:

- Controlled impact test [10,12–16];
- Compression test [17–20];
- Vibration test [21,22].

    As well as non-destructive techniques:

- Artificial vision systems [23];
- Hyperspectral imaging (HSI) [24–26];
- Visible and near-infrared spectroscopy (Vis-NIR) [14,27,28];
- Nuclear magnetic resonance [29–33];
- Thermal or ultrasound imaging [34–36];
- Electrical impedance [37].

In the same way, there are also numerous works focused on the quantification of different physical and geometric parameters (such as diameter, length, weight, volume) [7,38–42], or the quantification of the susceptibility to suffer damage by the fruit [10,13,43–46]. There are numerous efforts by researchers to find out what causes damage to the fruit and how it could be avoided or, in any case, reduced as much as possible, so that it does not affect its quality.

However, the main non-destructive techniques used by these authors either do not penetrate the fruit body and are limited only to characterizing the fruit surface (visible and near infrared spectroscopy, thermal imaging, or, recently, hyperspectral imaging), or use techniques that can penetrate the fruit and allow obtaining 2D and 3D images of the interior but are excessively expensive and are not available to any user (e.g., Magnetic Resonance Imaging). The aim of this work is to provide a new method of identification and characterization of the superficial damage produced in table olives that are generated throughout the entire production process, through the use of state-of-the-art and economically affordable 3D equipment. The proposed system is based on the digitization and 3D measurement of olives to determine geometric parameters that are very complicated to measure accurately through the previously mentioned methods, given their high two-dimensional component based on images.

## 2. Materials and Methods

In the present work, non-destructive methods using 3D digitalization were carried out. Furthermore, the study also applied a digital image analysis, as a usual bruise analysis methods, to compare results.

## 2.1. Hardware

The Digital Image Analysis system was composed of a main DSLR camera (Nikon D7500, Nikon Corporation, Tokyo, Japan) with a mounted macro lens (Sigma 105 mm f/2.8 ex macro dg hsm, Sigma Corporation, Tokyo, Japan), connected to a laptop computer (MSI GT73VR 7RE Titan, Micro-Star International CO., LTD, Taipei, Taiwan) to trigger the shots. The system provides images of 5568 × 3712 pixels (20.9 Megapixel), 48 bits color depth, and 21.3 Mb/picture. The photographs were taken at an aperture setting of f/7.1, shutter speed E:1/13 s, light sensitivity ISO:100, focal length 105 mm macro, and exposure compensation +1 stop EV.

The entire system was stabilized on a tripod (Manfrotto 055 pro, Videndum Media Solutions Spa. Cassola, Italy). To avoid hard shadows, no flash was used. To prevent parasitic lights and reflections, the photographs were taken inside a softbox (Neewer, Shenshen, China). The lighting system consisted of two white LED spotlights (5000° K daylight color) on the outside and on both sides of the softbox. Before taking pictures, the monitor was calibrated using a ColorChecker Display calibrator (Calibrite LLC, Wilmington, NC, USA), and a color profile was generated using ColorChecker passport (Calibrite LLC, Wilmington, NC, USA). All photographs were acquired in Adobe 1998 color space and RAW format without compression to avoid loss of color information.

The acquisition of 3D models of the olives was carried out using a 3D scanner (Artec3D mod. Spider, Artec 3D, Senningerberg, Luxembourg). This is a handheld type of structured light technology scanner. It has a 3D point accuracy of 0.05 mm, enough to detect the small defects of the olive. In the same way, the 3D scanner was placed on a small tripod to avoid sudden movements during the scanning process. The olives were attached to a small stick and scanned on a turntable controlled from the laptop to achieve a semi-automatic system.

## 2.2. Samples

The sampling, taken from a farm with UTM coordinates 30 N 372041 4147700, consisted of the collection of 103 olives, of two different varieties (Hojiblanca and Picuda), of which, 60 were selected after a mechanical harvest and 43 were selected by hand on the tree, 13 of them showing some disease (as a consequence of the presence of *Camarosporium dalmaticum* L. or *Sphaeropsis dalmatica* L.) on the fruit. From the point of view of geometry, there was no difference between damaged and diseased olives, beyond the effects of the disease, which affects the olive's skin and the flesh. Sampling of diseased olives was only carried out manually, as an example of olives with damage of a different nature than those produced in the harvesting process. The proposed objective was to characterize the geometry of damages of a different nature. The sample distribution can be seen in Table 1.

**Table 1.** Sample distribution.

| Harvest | Hojiblanca | Picuda | Diseased Hojiblanca | Diseased Picuda |
|---|---|---|---|---|
| Mechanical | 30 | 30 | - | - |
| Manual | 15 | 15 | 7 | 6 |
| Total | 45 | 45 | 7 | 6 |

## 2.3. Image Analysis

The image analysis was performed by the Fiji ImageJ2 (ImageJ 1.53q, National Institutes of Health, USA) image processing package. Fiji was released as open source under the GNU General Public License builds on top of the ImageJ2 core. It was run on a desktop PC (intel i7, 64 Gb RAM, and Geforce RTX 1080 Ti graphic card). The process was carried out in the Visual Computing Department of Technological Centre of Metal–Mechanical and Transport.

The olives were photographed with the described system one by one. The olives were positioned on the photography area randomly. A total of 100 olives of two varieties

(Hojiblanca and Picuda) were photographed. The entire set of photographs was imported into the Adobe Lightroom software (Adobe, San Jose, CA, USA), without any treatment, for review and classification. For the digital image analysis process, the best quality photograph of the three was chosen.

The color of the fruits, even in fruits of the same species, can vary slightly depending on many factors, such as the maturity state. Because this segmentation method strongly depends on the color of each individual pixel, it is very sensitive to these changes [47]. For this reason, a first identification of the healthy area and the damaged area was carried out manually. In this operation, the RGB values that discriminated only two classes were identified: healthy area and damaged area. With the range of RGB values determined, both for healthy and damaged areas, a semi-automatic routine was developed for the processing of the olive sample set.

The first step of the workflow (Figures 1 and 2) was the calibration of the image to determine the pixel/mm equivalence, obtaining an average of 85 pixels/mm. In this way, all the parameters and measurements made on the image will be shown as real metric units. Then, the ROI corresponding to the area of the image showing the olive was cropped. Damaged versus healthy areas were then segmented. To do this, the 16 bits/channel image (uncompressed.tiff image) was converted to Color RGB, with 32 bits per pixel, and the RGB channels were separated into individual channels: R, G, and B. The G channel allows discriminating with greater precision the healthy areas from the damaged ones. Two binary images were obtained, corresponding to whole olive area (total area binary image) and damaged area (bruise spots binary image). Thus, it was used as a basis for the mathematical operations of segmentation.

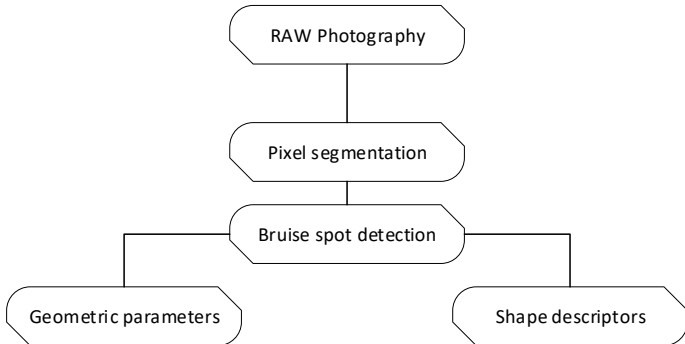

**Figure 1.** General image analysis workflow.

The semi-automatic routine was then run, in which the parameters to be identified in each of the detected damages had previously been indicated. In order to eliminate isolated pixels resulting from small variations in segmentation, a Median filter with a threshold value of 0.254 was applied to the image. Previously, the system was configured so that it did not consider those areas of less than 0.1 mm$^2$, because they could be artifacts resulting from the segmentation process. The bruised spots were automatically identified and measured.

The complete image analysis process required 8 min. Finally, the results were exported to a .csv file, in which each area corresponding to the damage, the total area of the olive, and a set of geometric parameters were identified. The measured parameters are (Table 2):

- Area, in mm$^2$: the average amount of pixels corresponding to spot defects or total olive area.
- X and Y coordinates of the center of the equivalent ellipse.
- Major and Minor, as primary and secondary axis of the best fitting ellipse to shape. This parameter can be assigned to the longest and shortest axes diameters of the olive in the photographs.
- Angle (0–180°), as the angle between the primary axis and a line parallel to the x-axis of the image.

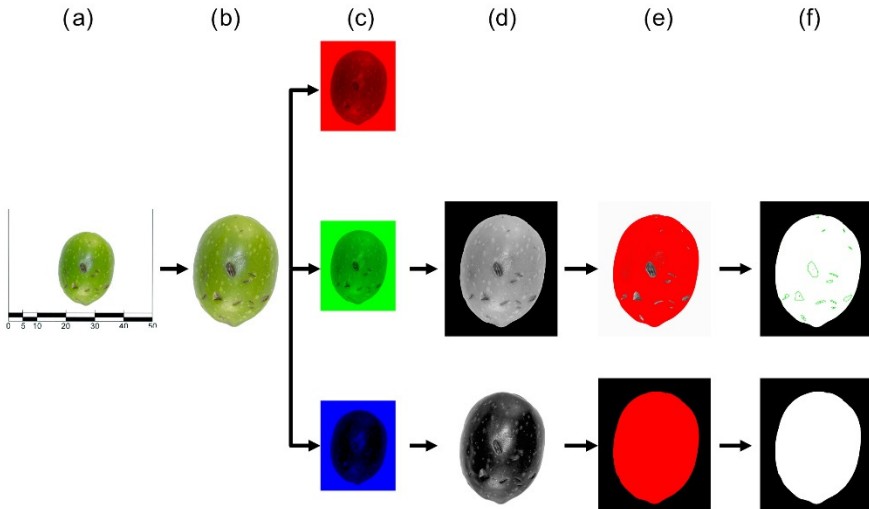

**Figure 2.** Bruise detection spot from image analysis workflow. Olive variety Hojiblanca: (**a**) original photographs, (**b**) ROI image cropped, (**c**) split color channels, (**d**) green (bruise spot) and blue (whole olive) channels for segmentation, (**e**) segmented images, and (**f**) results of image analysis.

**Table 2.** Example of results obtained from image analysis.

| Olive ID | Status | Harvesting | Total Area (TA) | Major Axis | Minor Axis | Angle | Circularity | Roundness | Aspect Ratio (RA) | Total Damaged (DA) | DA/TA (%) | Bruise Number |
|---|---|---|---|---|---|---|---|---|---|---|---|---|
| 10 | healthy | Pick handle | 412.867 | 26.322 | 19.971 | 93.211 | 0.859 | 0.759 | 1.318 | 0 | 0.000 | 0 |
| 40 | bruised | mechanic | 369.293 | 25.347 | 18.550 | 89.816 | 0.853 | 0.732 | 1.366 | 11.826 | 3.202 | 16 |
| 87 | bruised | mechanic | 344.234 | 26.198 | 16.730 | 83.989 | 0.821 | 0.639 | 1.566 | 79.121 | 22.985 | 28 |
| 100 | diseased | Pick handle | 321.447 | 25.623 | 15.973 | 92.461 | 0.796 | 0.623 | 1.604 | 13.973 | 4.347 | 1 |

In addition, several parameters, such as shape descriptors, were automatically calculated:

- Circularity (0–1), to analyze the closeness to a perfect circle of the particle (bruise area), where 1.0 is a perfect circle and 0 assimilates to an infinite elongated shape. Calculated as $4\pi * [\text{Area}]/[\text{Perimeter}]^2$.
- Aspect Ratio (AR), as a relation between major axis/minor axis. [Major Axis]/[Minor Axis].
- Roundness, to determine the degree of "sharpness" of the corners, both of the photographed surface of the entire olive and of the bruised areas, calculated as $4 * ([\text{Area}]/\pi * [\text{Major Axis}]^2)$.
- Number of spots: the number of bruised defects detected by digital image analysis.

These parameters will be used to analyze the possible origin of the damage of the olives based on their morphology (sticks, edges, pebbles, among others). Thus, one of the targets of this work is to try to determine the possible origin of the bruises of the olives. In this way, elongated damage can be related, for example, to sticks, and cuts with sharp edges or rounded bruise can be assigned to pebbles or machinery.

*2.4. 3D Scanning*

- The scanning of the olives was also carried out one by one. For this, Artec Studio 13 software (Artec 3D, Senningerberg, Luxembourg) was used. This software was also used to process the scans. The scheme of the scanning process was generally as described below (Figures 3 and 4).
- Three partial scans were performed per olive. Once the scans were finished, they were processed and the registration of the three scans was carried out to unite them into a single final scan.
- The next step was to remove the erroneous and outlier's points and generate the 3D polygonal mesh. For this, a resolution of 0.05 mm was established (Table 3).

- Finally, the 3D model of the olives was texturized (colored), and the results were exported in wavefront.obj format. The total time used in the entire process was 10 min per olive, time that can be significantly reduced in the case of scanning several olives at the same time.

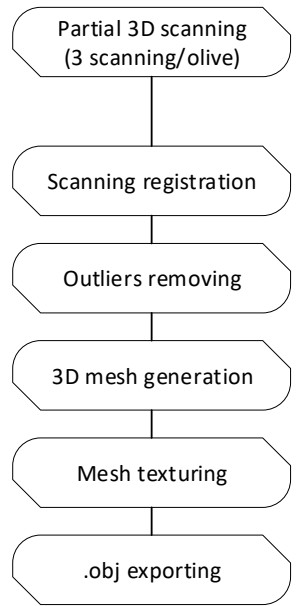

**Figure 3.** General scanning workflow.

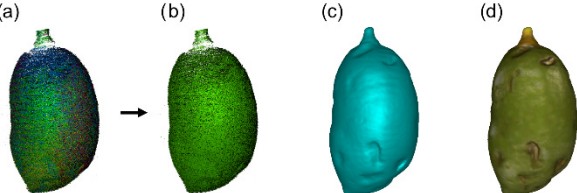

**Figure 4.** 3D scanning process: (**a**) partial scans; (**b**) scans registering; (**c**) 3D mesh generation, and (**d**) 3D mesh texturing.

**Table 3.** Example of results obtained from scanning workflow.

| Olive ID | Status | Harvesting | Resolution (mm) | File Size (Mb) | Polygon number | Volume (mm³) | Surface (mm²) |
|---|---|---|---|---|---|---|---|
| 7 | healthy | Pick handle | 0.05 | 7.82 | 126,142 | 93.211 | 0.859 |
| 32 | bruised | mechanic | 0.05 | 4.70 | 76,464 | 89.816 | 0.853 |
| 65 | bruised | mechanic | 0.05 | 4.27 | 69,572 | 83.989 | 0.821 |
| 98 | diseased | Pick handle | 0.05 | 3.76 | 61,596 | 92.461 | 0.796 |

The analysis of the olives made with the high-resolution 3D models was carried out in four stages:

- Curvature map of the mesh. The first step was to automatically detect the damage areas, such as those areas that were far from the ideal olive without damage. On the 3D olive meshes (Figure 5a), the modifications in the curvature of the surface are visualized, because not all the defects that the olives present have color variation to allow them to be analyzed; this is a problem of the image analysis that the three-dimensional analysis does not present. The texturing of the 3D models allows one to verify the goodness of the defects obtained by the three-dimensional analysis and

compare them to the image analysis. To represent these variations in the curvature of the olive surfaces, a zebra analysis, a system widely used in product design, has been tested but does not clearly represent the damage, as can be seen in Figure 5b. Instead, a model with a color scale that marks the intensity of the surface deviations was opted for: Figure 5c,d.

- ▪ Creation of 3D virtual olive without damage. To calculate the bruising, an ideal olive without defects, called a "virtual olive", is needed. Starting from the 3D mesh, (Figure 6a), and based on the curvature of the mesh (Figure 6b), the defects captured by the 3D scanner will be verified, hiding the photographic texture (Figure 6c). In this way, the 3D mesh of the virtual olive with an ideal surface is created. For this, the Polyworks Suite metrology, release 2012 software (InnovMetric Software Inc., Quebec, Canada) was used. The operation creates a theoretical surface without defects directly from the polygonal model. The theoretical surface and the olive 3D polygonal model are then compared in order to measure surface shape deviations. This allows the detection of some surface defects that could not have been detected by using usual color maps. The maximum defect width was imposed to 10 mm. The tolerance limits to the color map were ranged between 0 (normal olive) to –1.2 mm (max. bruise depth) (Figure 6d).

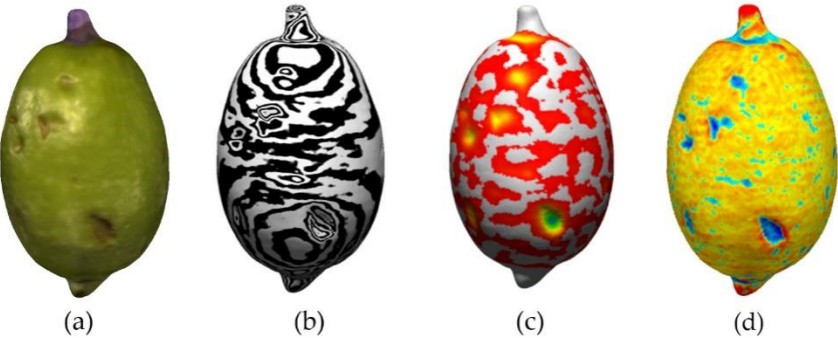

**Figure 5.** Curvature map of the mesh: (**a**) 3D mesh texturing; (**b**) zebra curvature; (**c**) curvature 1, and (**d**) curvature 2.

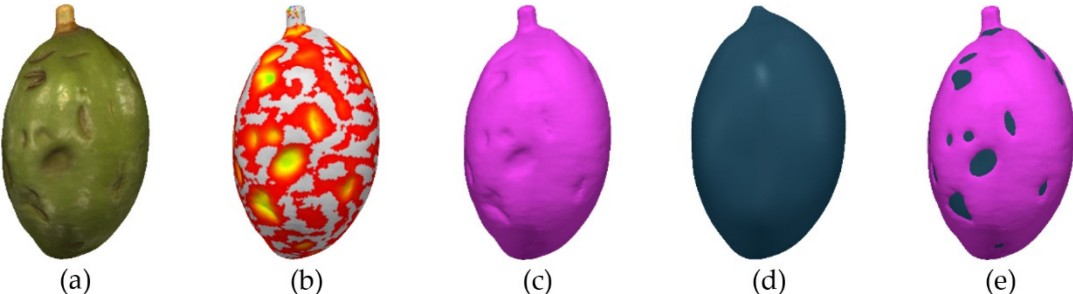

**Figure 6.** Creation of 3D virtual olive: (**a**) 3D mesh texturing; (**b**) curvature; (**c**) 3D mesh generation; (**d**) virtual olive, and (**e**) superposition of (**c**,**e**).

An overlay of the original mesh without texture (Figure 6c) and the virtual olive mesh, (Figure 6e), visually shows us the damage to the olive.

- ▪ NURBS model. In order to carry out an analysis of the damage, the value of the surfaces and volumes must be obtained, performing as a previous step the conversion of the 3D meshes into parametric models that allow us to perform mathematical operations.

The mathematical solids obtained are formed by NURBS (Non-Uniform Rational B-Splines) surfaces. For the creation of these models, it will be necessary to eliminate any defect in the 3D meshes. Different fruits have been modeled using NURBS models [48].

Geomagic Design X Build version 2016 software (3D Systems, Inc., Rock Hill, South Carol., USA), for reverse engineering, allows one to select between mechanical and organic meshes and different options, as summarized in Table 4.

**Table 4.** NURBS parameters.

| Mesh | Organic |
|---|---|
| Patch count | 200 |
| Fitting method | Adaptable |
| Setting options | Max. geometry capturing and tolerance 0.015 mm |

The options shown allow one to create exactly the original mesh of the olives, starting with the models of the real olives (Figure 7a) and later creating the virtual olive (Figure 7b).

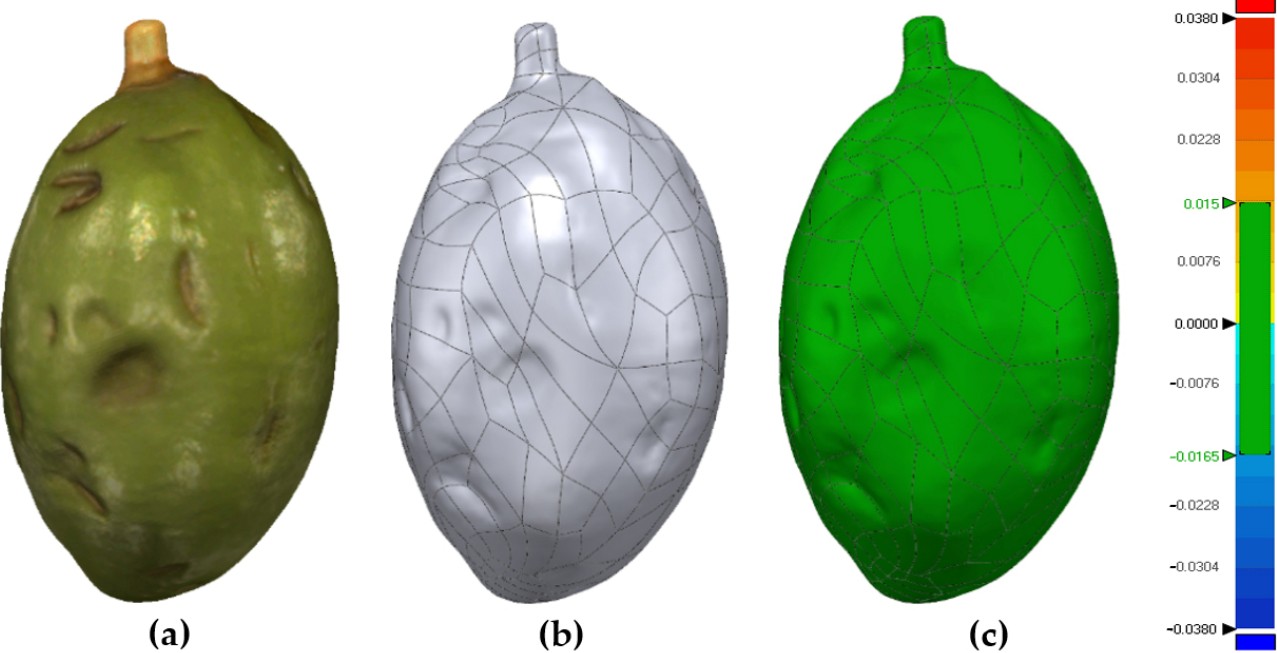

**Figure 7.** NURBS model of real olive: (**a**) 3D mesh texturing; (**b**) NURBS surfaces, and (**c**) surfaces deviation.

To verify the capture of the details in the NURBS surfaces, the deviation of the original mesh is compared with the mathematical mesh, representing in green all the surfaces that present a deviation of less than 0.015 mm; see Figure 7c.

The same procedure is repeated for the virtual olive, verifying that the deviations are within the imposed limit of 0.015 mm; see Figure 8c.

- Calculation of solids and surfaces. Obtaining the solids formed by NURBS surfaces allows the realization of Boolean operations, with which the solids that complete the defects of the real olives are obtained. Figure 9a represents the real olive, Figure 9b the virtual olive, and Figure 9c the superimposition of the two solids.

To perform the Boolean subtraction operation, the solid of the virtual olive (Figure 9b) is taken as the target object, and the solid of the real olive (Figure 9c) is used as the cutting element, obtaining the solids of Figure 10.

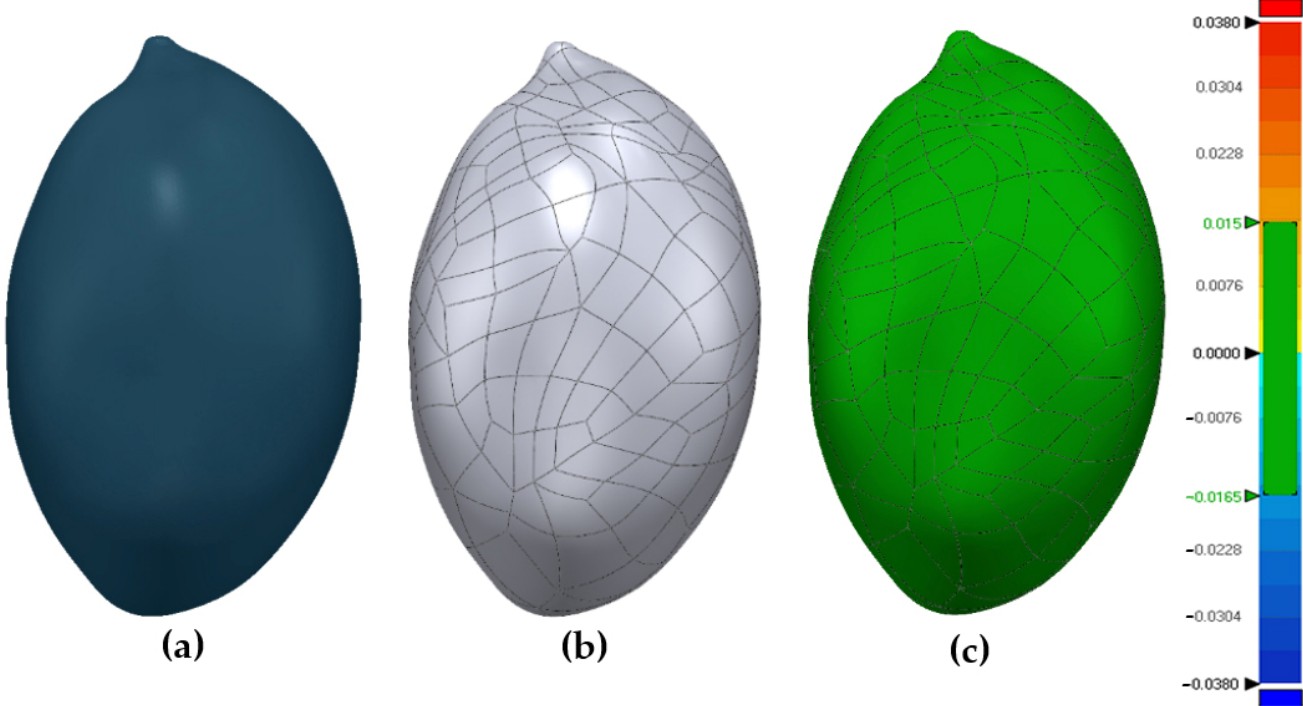

**Figure 8.** NURBS model of virtual olive: (**a**) virtual olive; (**b**) NURBS surfaces, and (**c**) surfaces deviation.

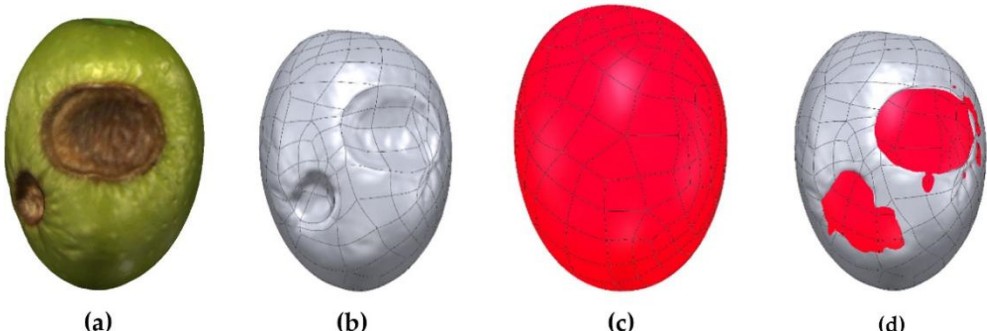

**Figure 9.** NURBS solid overlay: (**a**) 3D mesh texturing; (**b**) NURBS surfaces real olive; (**c**) NURBS surfaces virtual olive, and (**d**) superposition of (**b**,**c**).

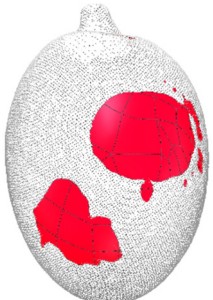

**Figure 10.** Boolean operation result.

These 3D solids represent the volume of each of the area defects, Figure 11.

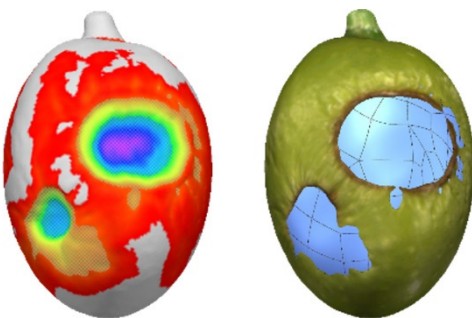

**Figure 11.** Diseased volumes.

From these 3D solids (Figure 11), the exterior surfaces (Figure 12a) and interior (Figure 12b) of each imperfection are extracted.

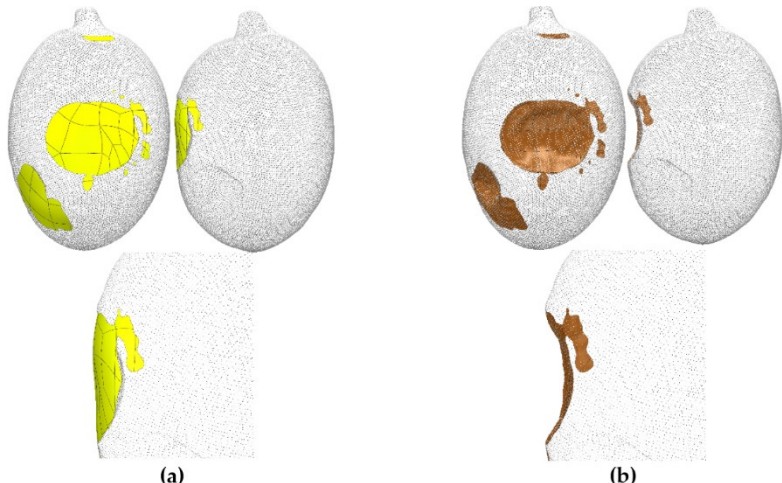

**Figure 12.** Diseased areas: (**a**) 3D outer surface, and (**b**) 3D inner surface.

After performing all the operations, all the imperfection data are extracted, both for solids and for external and internal surfaces, through a 3-min routine.

## 3. Results

In this work, two techniques for the characterization of damage in table olives have been used. Digital image analysis (DIA), as a relatively cheap, affordable, and well tested method, was used to compare the results with the proposed method based on the 3D models of olives. In recent years, more advanced results have been obtained through digital image analysis by using the visible and infrared spectrum, thermal imaging, etc. The main problem that all of them show is that they are based on a 2D model of the olive. In this aspect, the measurements obtained are partial and may be deformed towards the boundaries of the image due to the round geometry of the olives (Figure 13). Recently, Sola-Guirado et al. [49] provided a solution to this problem by partially segmenting the area without deformations of the image on which the defects are measured. These authors took 24 pictures per fruit and then cropped and merged them in a linear patter. Thus, the time to obtain and process the image is also multiplied by 24. The 3D techniques permit us to obtain, in a short time, a 3D real model of an olive, without deformations and with high-resolution details (up to 50 microns).

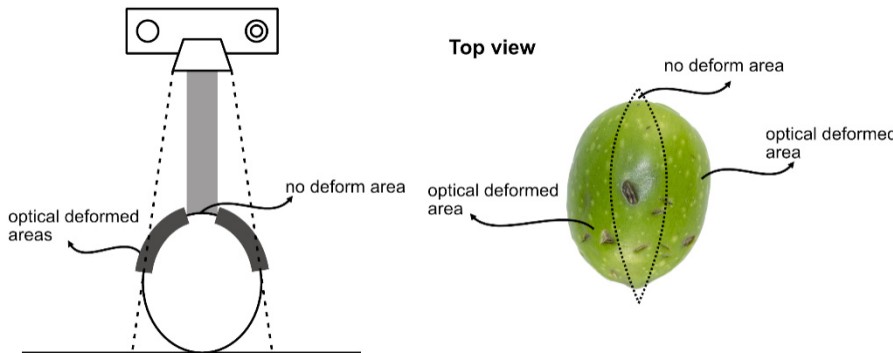

**Figure 13.** Optical deformation scheme affecting bruise spot measurements.

*3.1. Digital Image Analysis*

The main objective of this study was to obtain the most relevant parameters that would allow characterizing the damage to the olives. In recent years, more detailed studies have been published that delve deeper into these imaging techniques. These parameters will be compared later with the 3D equivalents of the olive models to validate the accuracy of the proposed method.

Firstly, it has been observed that the total area calculated from the image analysis shows a distribution close to normal, with a certain bias to the right. In general, the area from image analysis results is greater in the Hojiblanca variety than in the Picuda variety, with average values that range between 320 and 380 mm$^2$ for the Picuda and between 360 and 440 mm$^2$ for the Hojiblanca (Figure 14). From these results, it is also observed that the size and number of the damages is not related to the size of the fruit.

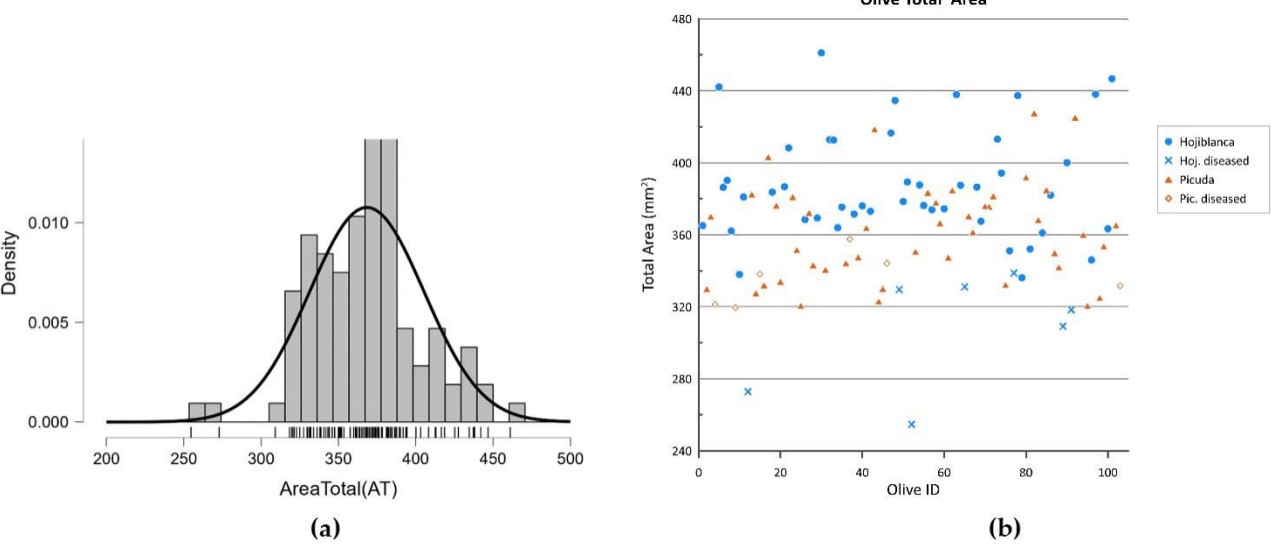

**Figure 14.** Total area distribution: (**a**) histogram vs. theoretical probability density function (PDF), (**b**) scatter plot.

In the same way, the intrinsic parameters of each variety (circularity, Feret's diameters, roundness, total area, among others) are also related to the olive's variety. In the case of circularity and roundness, they are both higher in the Hojiblanca variety (Figure 15). On the other hand, the determined aspect ratio (AR) is higher in the Picuda variety, showing that this variety has a more elongated geometry than the Hojiblanca. In this variety, a greater dispersion of the AR values is also observed (AR 1.5–1.75), while for Hojiblanca olives the range is much more limited (AR 1.3–1.4) (Figure 16). For these three shape descriptors, the distributions are bimodal, showing a clear differentiation between both types of varieties.

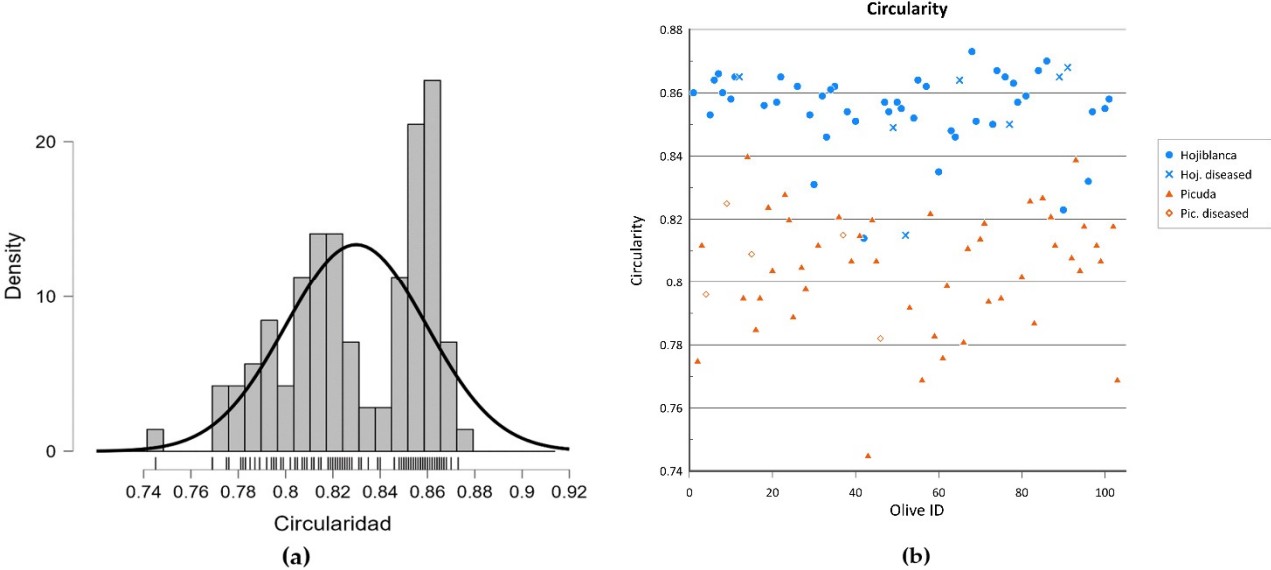

**Figure 15.** Circularity distribution: (**a**) histogram vs. theoretical probability density function (PDF), (**b**) scatter plot.

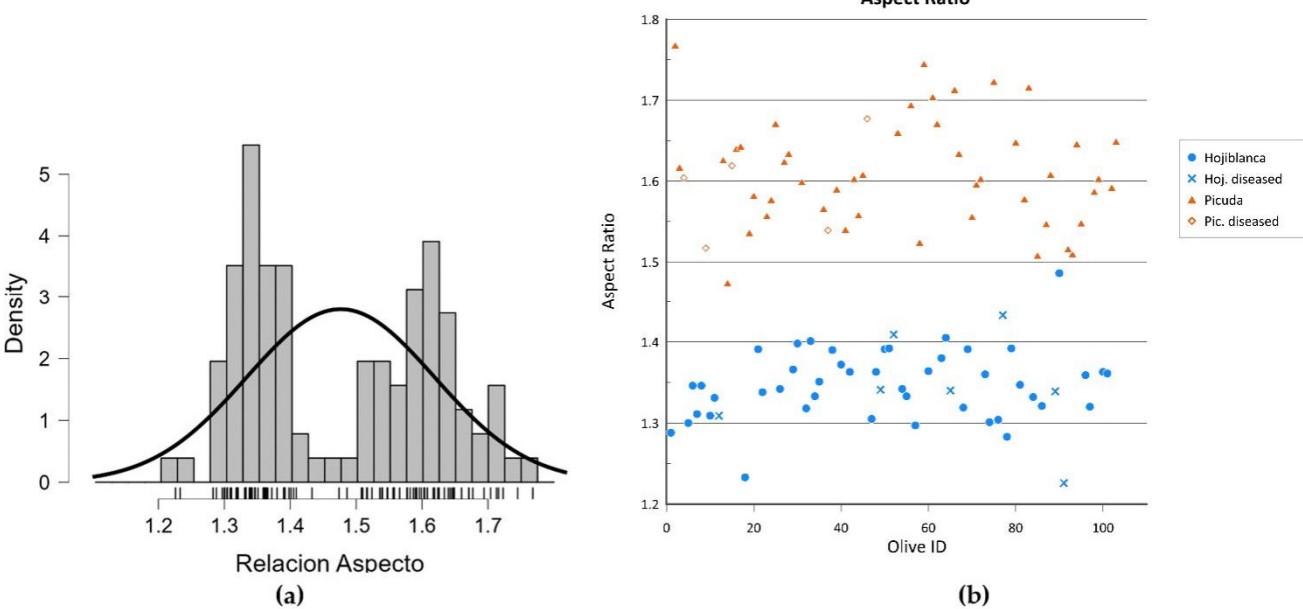

**Figure 16.** Aspect Ratio distribution: (**a**) histogram vs. theoretical probability density function (PDF), (**b**) scatter plot.

Regarding the number of damages, the image analysis does not show that one variety is clearly more damaged than the other, although, the highest values are observed in the Picuda variety. This is also corroborated, not only by the number of individual damages detected, but also by the relationship between the damaged area versus the healthy area. Once again, the olives with the highest rate of damage observed are the Picuda variety (Figure 17).

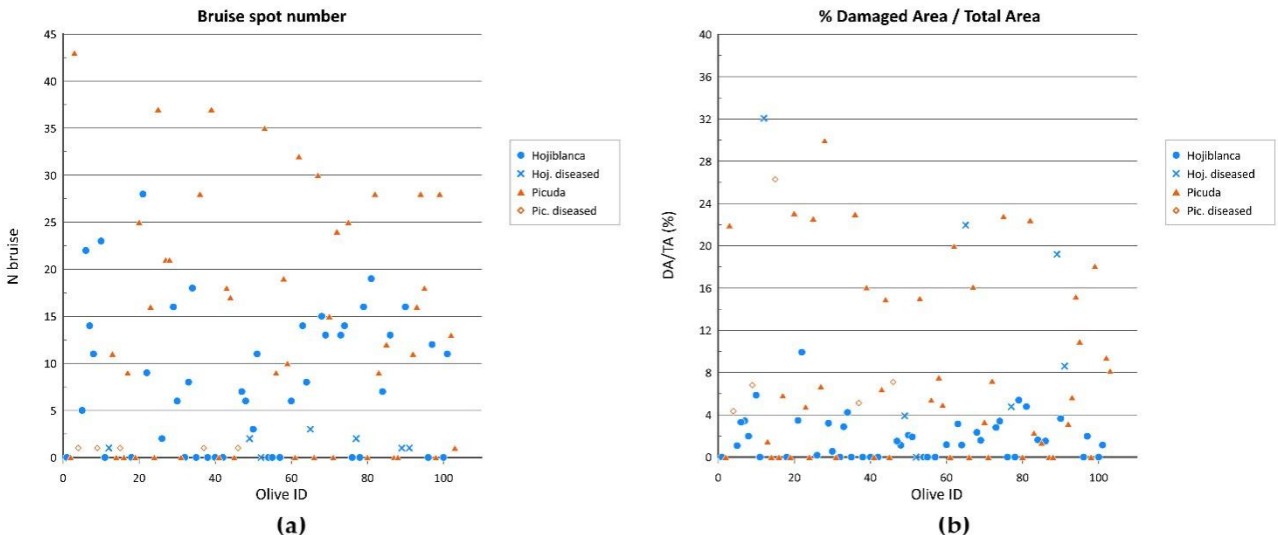

**Figure 17.** (**a**) Number of spot bruised distribution, (**b**) Percentage of damaged area versus total area.

In the case of the analysis of the identified damage, the same formal parameters have been calculated. From the image analysis, a total of 1028 damages (spot bruises) have been detected for all the olives studied. From the results obtained, it is observed that:

(1) There is no direct relationship between the area of damage and the variety of olive. The highest percentage of damage size is in the range of 0.1–10 mm$^2$, with the highest surface values corresponding to diseased olives (Figure 18a).

(2) Parameters such as circularity or roundness do not allow discrimination between varieties of olives, both varieties being due to damage with similar geometries. In the same way, there is no differentiation between damage morphology that allows grouping damage to a specific cause (sticks, stones, machinery, among others) (Figure 18b)

(3) Regarding the aspect ratio (AR) of the damage, the vast majority of damage is in the range of 1–4, denoting a relatively circular geometry. Some of the damages with values greater than 5–6 correspond to elongated damage and are assigned to blows and cuts produced by elements with a high linear component, such as sticks or elongated sharp areas. (Figure 18c).

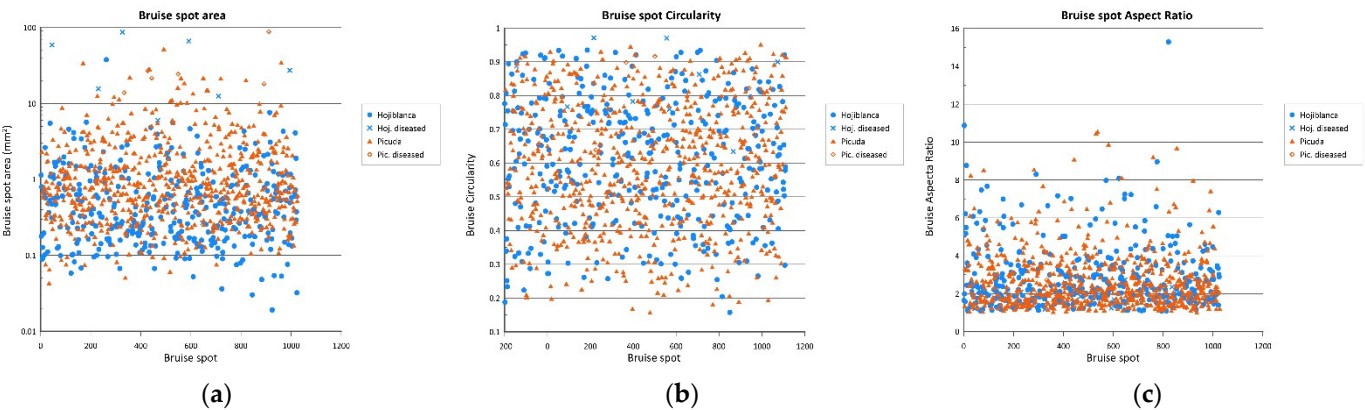

**Figure 18.** (**a**) Area of spot bruised distribution, (**b**) Bruise spot circularity distribution, (**c**) Bruise spot Aspect Ratio.

### 3.2. Three-Dimensional Analysis Results

The first results correspond to the meshes resulting from the 3D scans. In a visual analysis of the meshes, the different types of disturbance can be appreciated (See Figures 19 and 20).

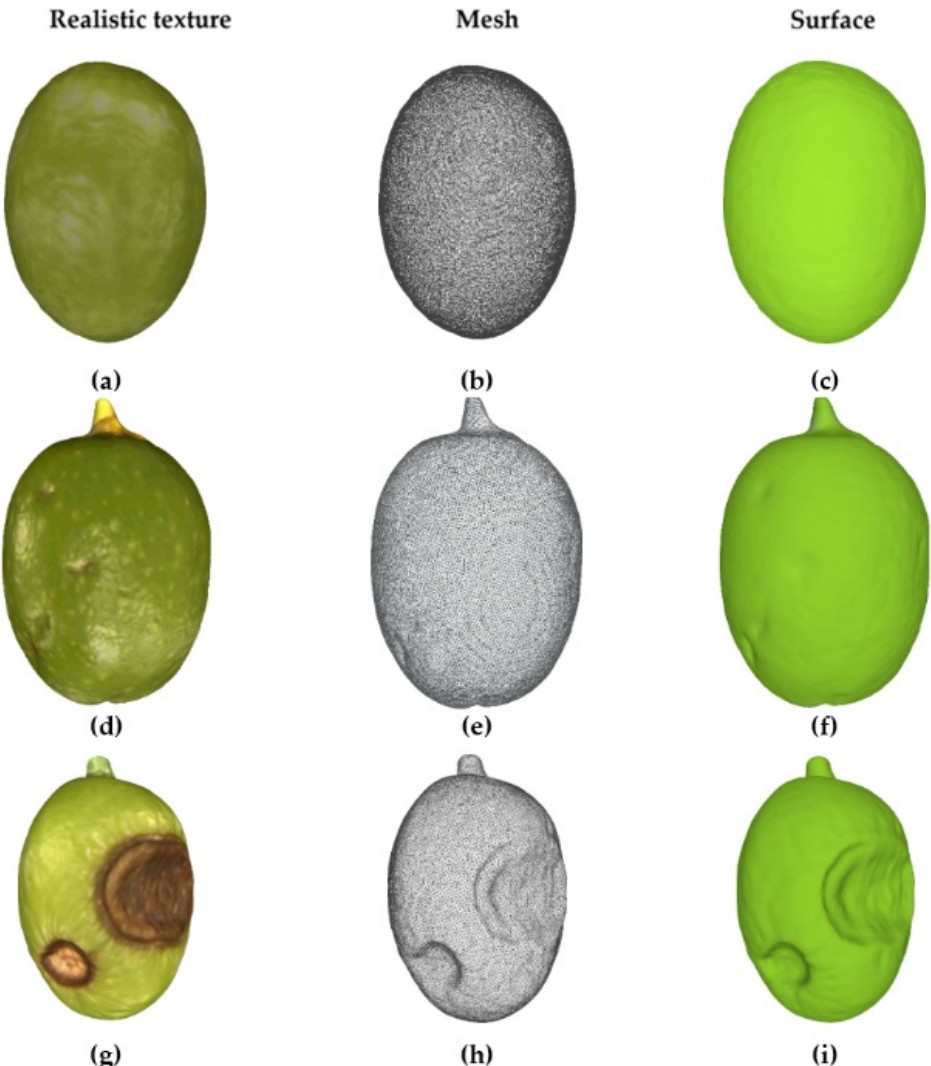

**Figure 19.** Examples of Hojiblanca variety olives: (**a**,**d**,**g**) 3D mesh texturing; (**b**,**e**,**h**) mesh, and (**c**,**f**,**i**) surface.

The order in which the data have been distributed is shown in Table 5.

**Table 5.** Data distribution.

| Harvest | Hojiblanca | Picuda | Diseased Hojiblanca | Diseased Picuda |
|---|---|---|---|---|
| Mechanical | 1–30 | 31–60 | - | - |
| Manual | 61–75 | 76–90 | 91–97 | 98–103 |

Starting from the models of the virtual olives, the surface data and reference volumes are obtained. Figure 21 shows the distribution of the volumes of the olives, with an average of 3130.27 mm$^3$.

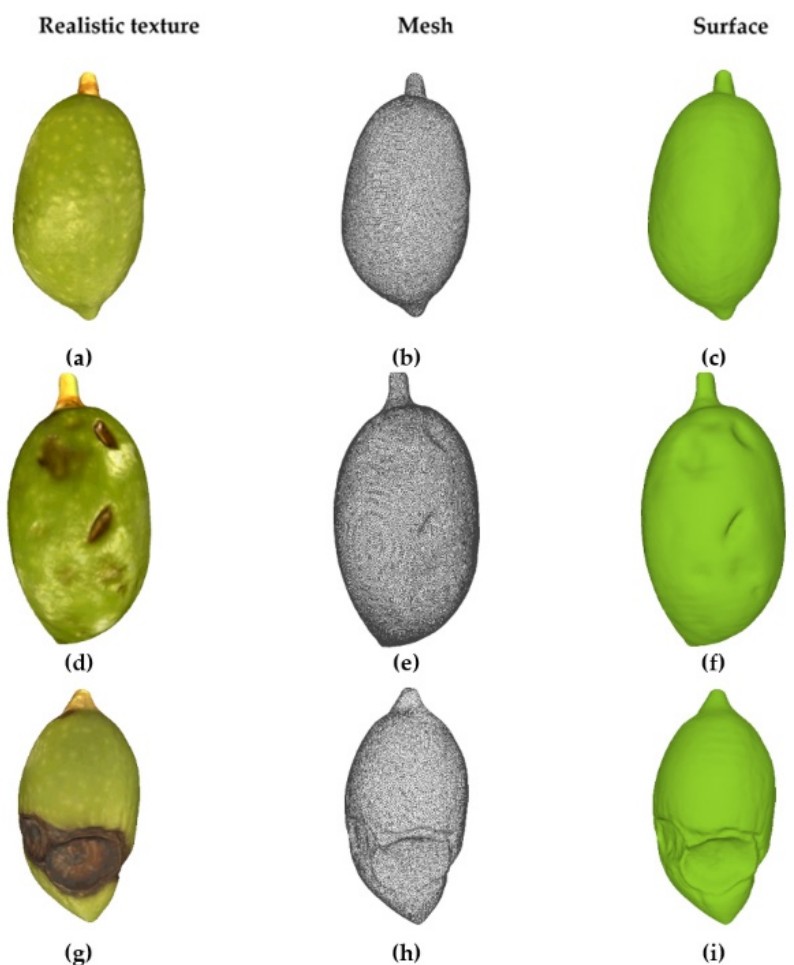

**Figure 20.** Examples of Picudas variety olives: (**a**,**d**,**g**) 3D mesh texturing; (**b**,**e**,**h**) mesh, and (**c**,**f**,**i**) surface.

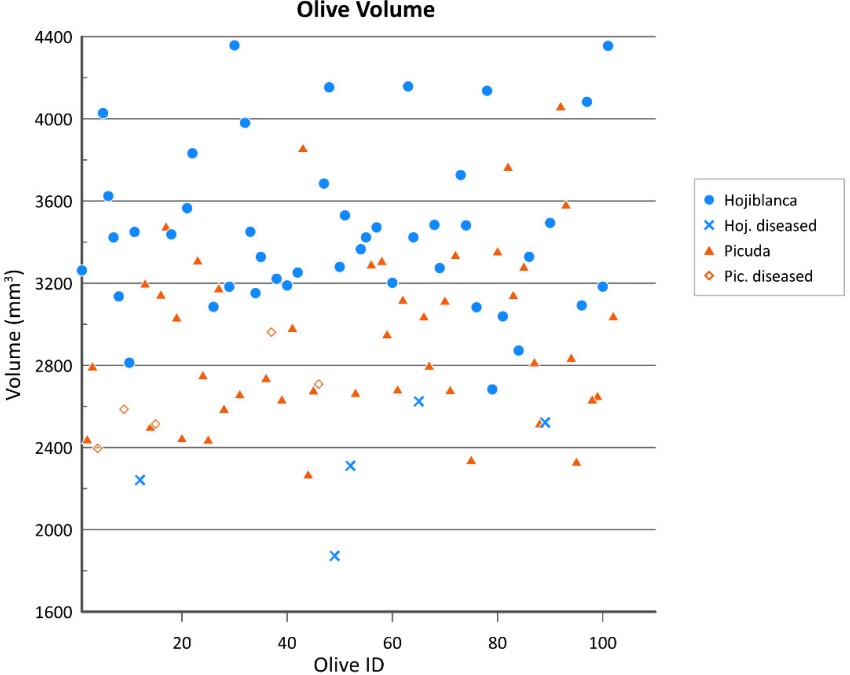

**Figure 21.** Total Volume plot.

The surfaces of the olives are represented in Figure 22.

**Olive Total Surface**

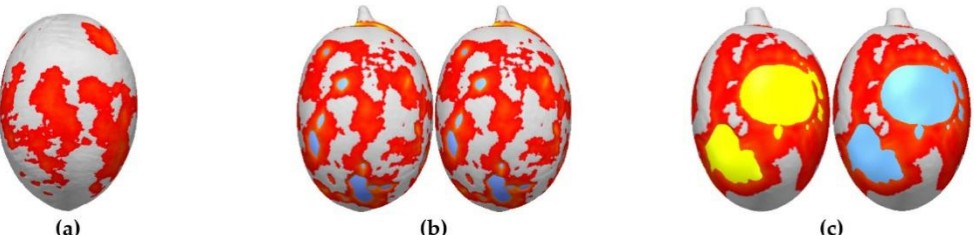

**Figure 22.** Total Area from 3D plot.

Applying the methodology described in Materials and Methods, the 103 samples of olives were analyzed, obtaining the models of imperfections that the real olive presents. The difference in the imperfections obtained can be seen in the examples of Figures 23 and 24, being (a) the olives with manual harvesting, (b) the olives with mechanical harvesting, and (c) the diseased olives.

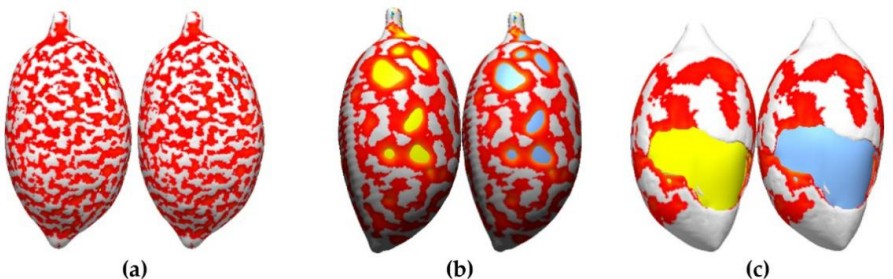

(a)     (b)     (c)

**Figure 23.** Damage to Hojiblanca variety olives: (**a**) the olives with manual harvesting, (**b**) the olives with mechanical harvesting, and (**c**) the diseased olives.

(a)     (b)     (c)

**Figure 24.** Damage to Picuda variety olives: (**a**) the olives with manual harvesting, (**b**) the olives with mechanical harvesting, and (**c**) the diseased olives.

The database was completed with the values extracted from the solids and from the external and internal surfaces of the imperfections. The analysis of the results was differentiated into two parts: surface analysis and solid analysis.

### 3.2.1. 3D Solid Analysis

It can be seen that the classification of the olives by the surface of the solids, Figure 25, shows a distribution according to the olives that is consistent with the damage of each type of olive. To complete the classification, the distribution of the damage volumes was calculated, which provided a more precise distinction between the internal affectation of the olive pulp, obtaining a less dispersed classification, Figure 26.

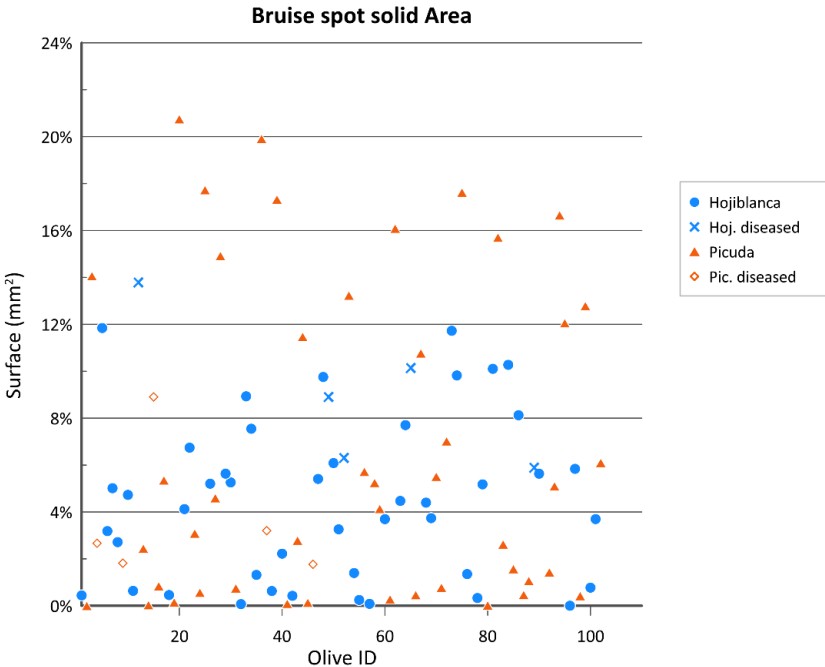

**Figure 25.** Bruise spot solid area plot.

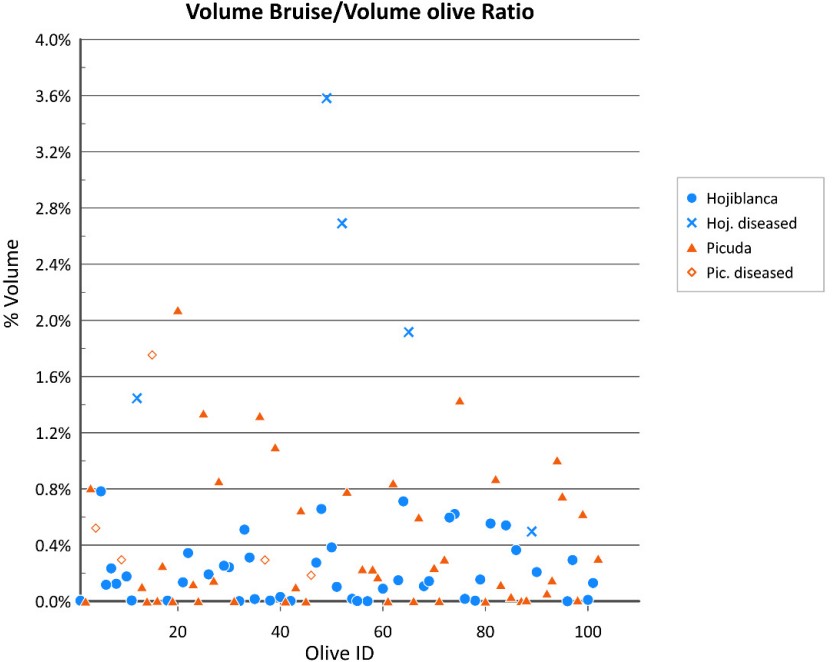

**Figure 26.** Volume bruise—olive volume ratio plot.

### 3.2.2. 3D Surface Analysis

Initially, the hypothesis of being able to differentiate the damages by comparing the two external and internal surfaces of the imperfections was raised. Figure 27 shows the percentage of damage compared to the total surface of the olive on the outer surfaces. These surfaces practically overlap, so they can be used interchangeably.

**Figure 27.** Internal and external bruise area plot.

### 3.3. Comparison of Parameters for the Estimation of Fruit Bruising

The total damaged surface and volume calculated using the 3D analysis with solid or digital image analysis are shown in Table 6. The 3D analysis can provide a qualitative value because it allows the calculation of the damage depth. It can already be seen that the digital image analysis method reports much lower values than the 3D method, because, in addition to losing information from different parts of the olive, the true magnitude of the damage in real projection is not appreciated. The damage surface estimation through image analysis is only slightly accurate in comparison with 3D analysis, when the damage remains in the area without deformation (Figure 13), as can be observed in Figure 28a. However, if the damage is in the area affected by the deformation, the estimated damage does not match the real one (Figure 28b). However, a very good relationship between the area determined with both methods has been determined, i.e., the area determined with the image analysis (one single photo) is of the order of 2.98 times smaller than the damaged area calculated with the 3D method. This can be seen in Figure 29, where the 2D area values have been multiplied by this value to obtain an indication of the surface from 2D compensated, which is highly correlated with the area obtained with the 3D method.

**Table 6.** Fruit bruise index with different methods. Values shown are mean $\pm$ standard deviation.

| Fruit Status | Bruise Index * (%) from 3D Analysis | Bruise index ** (%) from Digital Surface Analysis |
|---|---|---|
| healthy | $0.5 \pm 0.5$ | $0.0 \pm 0.0$ |
| bruised | $8.2 \pm 5.1$ | $7.5 \pm 7.4$ |
| diseased | $6.6 \pm 3.7$ | $12.9 \pm 9.5$ |

* Calculated as the internal 3D surface of damage and the 3D surface of fruit ratio; ** Calculated as the 2D area of damage and 2D area of fruit ratio.

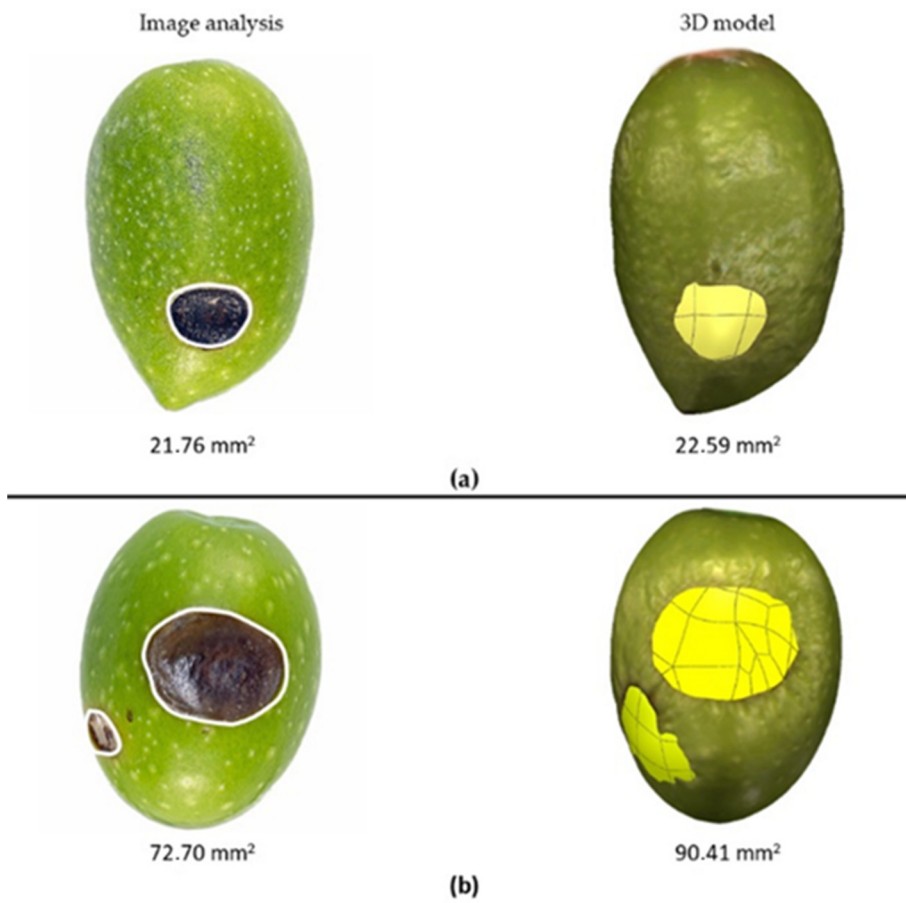

**Figure 28.** Bruise area calculated by image analysis vs. 3D analysis, with damages in the non-deformed position (**a**) and in the deformed position (**b**) regarding the top horizontal plane.

### 3D-2D compensated Bruise spot Areas Comparison

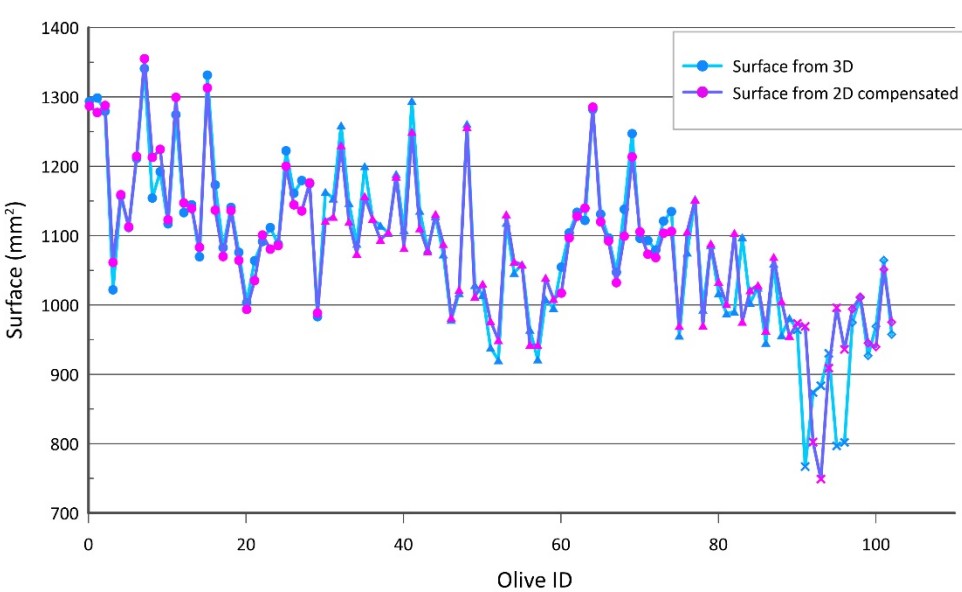

**Figure 29.** Surface of damage calculated from 3D and 2D image digital analysis multiplied by a 2.93 value.

However, this ratio between the areas calculated with both methods does not follow this ratio when the total fruit area of either a 2D face or the total 3D ellipsoid is taken into

account. Table 6 shows a summary of the main bruise index, in percentage, obtained to characterize the damage to the fruit from the geometric parameters calculated with the different techniques. The results show that there are significant differences (*t*-student test, $p < 0.05$) between the analysis methods for the diseased fruit, but there are not significant differences for bruised fruit.

Regarding the damage classification and their status, there were significant differences (ANOVA, Duncan's test, $p < 0.05$) for the three parameters (Table 7) between damaged olives and healthy olives. The mean damage in diseased olives is higher than in mechanically damaged olives, but not significantly different due to the high deviation between samples.

**Table 7.** Parameters calculated for characterization of the fruit bruising. Values shown are mean $\pm$ standard deviation. The same letter in the same column indicate no significant difference between fruit status (Duncan's test, $p < 0.05$).

| Fruit Status | Total Area (mm$^2$) of Damages from . . . | | Total Volume (mm$^3$) of Damages from . . . |
|---|---|---|---|
| | 3D Analysis | Digital Image Analysis | 3D Analysis |
| healthy | $11.8 \pm 11.2$ A | $0.2 \pm 0.2$ A | $0 \pm 0$ A |
| bruised | $176.1 \pm 102.3$ B | $27.6 \pm 26.8$ B | $13.7 \pm 10.8$ A |
| diseased | $113.1 \pm 61.8$ B | $40.8 \pm 28.6$ B | $31.2 \pm 22.9$ A |

## 4. Discussion

The field of image processing has been the subject of intensive research and development activities for several decades. Rapid technological advances, especially in terms of computing power and network transmission bandwidth, have resulted in many remarkable and successful applications. Current techniques for defect detection are based on 2D technologies; these technologies are efficient and widely implemented in the industry. This broad area encompasses topics such as image/video processing; image/video analysis; image/video communications; image/video sensing, modelling, and representation; computational imaging; electronic imaging; information forensics and security; medical imaging; and machine learning, which have all been applied to these respective topics. In recent years, one of the fields of study of image analysis has been the detection and characterization of the damage produced in the fruit due to different causes. However, the image analysis technology presents technical limitations, derived mainly from this two-dimensional character:

1.  When studying objects with a rounded three-dimensional geometry (spherical, cylindrical, ellipsoidal, etc.), a single photograph does not capture the entire object. In this case, it is necessary to take serial photographs throughout the entire object. With the increase in the number of photographs, it must be added that only a small part of the resulting photograph presents a minimum optical deformation. In this sense, the correct measurement of defects is only accurate in a small part of the fruit. Measuring damage to the entire fruit is therefore time consuming. In this work, a single capture per olive has been carried out as the most common method of capture. According to the results, in this single capture, only one third of the surface of the olive was correctly registered.

2.  Image analysis techniques, with few exceptions, are not penetrating. In this sense, the only parameters that can be obtained are the references to the outer surface of the olive, without being able to analyze the development of the damage that affects the pulp. To study the geometry of the damage towards the interior of the fruit, it is necessary to use other complementary techniques.

3.  As usual, in any technique based on color images, it is very dependent on the photographic capture conditions (camera, lens, triggering parameter, etc.). For large olive samples, light conditions must be maintained. Thus, variations in these conditions result in variations in the color of the olive surface.

4. In this sense, throughout the semi-automatic process it has been observed that some color variations not corresponding to damage have been wrongly assigned to damage. The routine has not been able to correctly segment the pixels. To avoid these errors, manual segmentation must be resorted to, leading to an increase in processing time.

5. Finally, the quality of the image in terms of pixels (resolution) will imply a better quality in the image analysis calibration. This process is fundamental for an optimal measurement of the parameters of the olive.

The three-dimensional analysis carried out avoids the problems described above, because complete information is available on the entire olive, which allows us to carry out a more exhaustive analysis of defects and more precise classifications. The weak points of the system can be highlighted:

1. Slightly slower process compared to image analysis. High-resolution 3D scanning requires a precise capture of the olive. This process is 5 min slower than a simple photograph, but the result is the 3D model of the complete olive, without deformations derived from optics and with results that do not depend on environmental conditions. In the case of serial photographs to obtain the olive, the multiplication process has to be completed, so that time difference can be reduced.

2. As with any technique, 3D digitization requires specific equipment. High resolution 3D scanners are not cheap. However, they always maintain the same capture conditions, speed and results. Another advantage is that the results are metrically definitive, avoiding calibration errors that in small sizes can lead to large differences in measurement. The capture of color images by 3D scanners helps to characterize the damage. Thus, with a single piece of equipment it is possible to obtain measurements of both color and shape.

3. For the use of this technology, a higher qualification of the operators is necessary.

The three-dimensional methodology focused on the analysis of solids offers more precise results and a faster procedure. Because olives are an organic product that undergoes continuous degradation over time, 3D scanning provides a solid and immutable database over time, which allows us to carry out a subsequent analysis or verification of bruising.

From the three-dimensional analysis, the detection of damage (deformation or concavity) not visible in the image analysis is also observed, because there are no color changes on the surface of the olive. Thus, in healthy olives, three-dimensional analysis techniques allow detecting small defects that go unnoticed by image analysis.

In subsequent advances of the work, it is planned to implement artificial intelligence methods for the automatic characterization of damage and its assignment to the origin of the damage, which will allow the redesign of both harvesting processes and machinery [50,51].

## 5. Conclusions and Future Work

In this work, a new methodology has been developed for the characterization of damages (bruise and disease) in table olives. It uses 3D technologies as a complement to standard 2D analyses, such as digital image analysis. The 3D digitization of table olives through structural light scanners allows one to obtain a precise record of the damage of table olives in the whole olive. Through this method, the main shape parameters have been estimated, and these characterize both the olives and the damage observed in them in a more precise and real way, because errors derived from optical deformations are avoided. A clear advantage of the proposed methodology, compared to traditional methodologies, is the ability to analyze the affectation of the olive pulp, being a fundamental parameter for its commercialization. It is also a relatively cheap and portable method compared to other 3D techniques used in the literature, such as CT.

A conversion factor has been calculated between the two-dimensional surface measurements obtained from image analysis and the results of 3D scanning. You may notice that by applying a scale factor of $\times 2.94$ to the surface calculated by image analysis, the resulting area is very similar to that obtained from the three-dimensional shape analysis. This factor is valid for the generality of the observed damages. In the case of very deep

damage, this equivalence could not be fulfilled, because the real surface differs significantly from the geometric one.

Future research should address the need to improve damage characterization and classification with 3D image processing methods, e.g., deep learning with improved 3D sensing and mapping techniques. These techniques could increase the speed of the 3D methodology, which, in combination with the use of a new system for the simultaneous capture of several olives, will reduce the processing time, allowing equalization of the two methodologies.

Future works could focus on the creation of an index based on the 3D methodology that completes the current indices.

It is intended to replicate the three-dimensional analysis in different fruits and their varieties to verify the universality of the system for the detection of defects by means of surfaces and volumes. This method of measurement based on three-dimensional real geometry aims to complement the usual methods of characterization of shape, providing a third dimension to the measurements and approximating more precisely the real dimensions of the damages.

**Author Contributions:** Conceptualization, J.R. and R.R.S.-G.; methodology, R.G.-M., R.R.S.-G., and E.S.-L.; software, R.G.-M. and J.R.; validation, R.R.S.-G. and E.S.-L.; formal analysis, R.G.-M., R.E.H.-F., and J.R.; investigation, R.G.-M. and J.R.; resources, R.E.H.-F. and E.S.-L.; data curation, R.G.-M. and J.R.; writing—original draft preparation, R.G.-M. and J.R.; writing—review and editing, R.R.S.-G. and E.S.-L.; visualization, R.E.H.-F.; supervision, R.E.H.-F. and E.S.-L. All authors have read and agreed to the published version of the manuscript.

**Funding:** This research received no external funding.

**Institutional Review Board Statement:** Not applicable.

**Informed Consent Statement:** Not applicable.

**Data Availability Statement:** The datasets generated for this study are available on request to the corresponding author.

**Conflicts of Interest:** The authors declare no conflict of interest.

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
