# Peer review of "Postharvest Geometric Characterization of Table Olive Bruising from 3D Digitalization"

_agronomy, doi:10.3390/agronomy12112732_

Round 1

Reviewer 1 Report

Q1. In the abstract, the author described 2D technology too much, and failed to fully describe the specific methods and advantages of 3D digital analysis technology.

Q2. In the abstract, the 3D bruise index mentioned by the author is not clearly defined in the text of the paper. What does it include?

Q3. In section 2.2 Samples, lines 140-141, "The sampling has consisted of the collection of 100 olives, of two different varieties (Hojiblanca and Picuda), of which, 60 have been selected after a mechanical harvest and 40 have been selected by hand on the tree, …". There are some questions here, the author should specify what is the difference between the geometric characteristics of mechanically grazed olives and diseased olives? Samples of diseased olives that were mechanically grazed during mechanical harvesting should also be included in the overall sample.

Q4. What are the geometric characteristics of mechanically grazed olives and diseased olives? What is the basis for this determination?

Q5. In section 2.3 Image Analysis, line 210, "Roundness, to determine the degree of angularity of the bruised areas". This statement conflicts with Table 2 below. A healthy olive with ID 10 should not have the parameter of roundness, while the table shows that the roundness is 0.759. Next sentence, in lines 211: "Solidity, as a measurement of the overall concavity of the bruise area". According to the following content of the article, it can be found that solidity is not mentioned.

Q6. In section 2.3 Image Analysis, lines 214-215, "These parameters will be used to analyze the possible origin of the damage of the olives based on their morphology (sticks, edges, pebbles, among others) ". It is suggested that the author add relevant content to explain why these parameters can be used to analyze the possible origin of the damage of the olives based on their morphology.

Q7. In Table 2, the 4 olive varieties are not illustrated. In addition, it is necessary to claim how to set the ID of 100 olives in the experimental group. The IDs of the four olives in Table 2 are inconsistent with those in Table 3. Why some images like Fig.28 have horizontal numbers representing olive ID, while others like Fig.26 seem to be random?

Q8. In section 2.4 3D scanning, the models of real olives should be added in Figure 5 and Figure 9 for comparison. In this section, it is not clear whether the olive in the picture is bruised or caused by disease. With 3D digital technology, the difference between bruises and diseases on olive surface is not obvious, so it cannot provide accurate reference for reducing and preventing bruising during harvesting and processing.

Q9. From Fig.29 in section 3.3, it can be found that the accuracy of area obtained by 3D analysis and image analysis is different, especially in the area affected by deformation. It is suggested that the author add specific description here to explain.

Q10. The comparison results of the two technologies are described too much in the discussion in section 4, which needs to be simplified. At the same time, this paper lacks conclusions and prospects. The author summarized that 3D analysis is slightly slower than image analysis (lines 528-533). However according to the article, the processing time of analysis is only described in line 231-234 of section 2.4, which is without time comparison.

Q11. There are also some details in this article. For example, in section 2.3 Image Analysis of lines 198-200, "Major and Minor, as primary and secondary axis of the best fitting ellipse to shape. This parameter can be assigned to largest and smallest dimensions of the olive in the photographs". It is inappropriate to consider that the minor axis corresponds to the smallest dimensions of the olive in the pictures. In lines 240-241, there is a grammatical error in this sentence: "The analysis carried out with the high-resolution 3D models of the olives was carried out in 4 stages". Results of the Fig.14 (b), Fig.15 (b), Fig.16 (b), Fig.17 (a&b), the experimental data is inconsistent with the above description. The number of the tested olives in these figures exceed the 100 subjects, and the number of diseased Hojiblanca (7) is inconsistent with the data in Table 1 (5). 

Reviewer 2 Report

Substantive assessment
Interesting work on the protection of delicate fruits and vegetables during harvest and storage with the use of computer image analysis. Olives, especially green ones, are very susceptible to any mechanical action, hence the need to control their quality at individual technological stages during their extraction and processing. 3D methods seem to be very useful for imaging the surface and shape of olives. This is not a new method. 3D cameras are used in many production processes on technological lines, but also in the field, e.g. when spraying trees in an orchard. The problem here is the large number of objects to be quickly "seen" in 3D. If the presented system deals with it at a good level, it is worth recommending. I recommend the authors of the manuscript to be interested in artificial intelligence and neural networks, which are well suited to similar tasks. You can "teach" a system of correct dimensions and shapes of individual olive varieties, which then works faster and more accurately.
As usual, in such works, I recommend that you also be interested in reliability and durability for newly developed complex structures. Examples of useful methods for quantifying the reliability of technical facilities in the field can also be found in the works of the MDPI publishing house, e.g.

 https://doi.org/10.3390/agronomy12061364

 https://doi.org/10.3390/ma14227014

MDPI publications are worth and should be cited, because the authors benefit from it (higher IF).
There is "chaos" in Figures 25 and 26, it is difficult to find meaningful relationships (no trend line and R2 coefficient). The same remark applies to figure 21 and 22 and figure 17 (poor quality)
in Figure 25, the ordinate seems to be incorrectly described. In turn, in Figure 26 the ordinate axis is 2 described by%. On the other hand, the abscissa axis has no description.

Editorial evaluation
• remove redundant "empty" lines, e.g. 60, 67 or 75 ...
• one page can be "saved" by placing References directly below the body of the text
• to specify the magnification, eg in line 553 we use the special multiplication sign  and not the letter x.

Reviewer 3 Report

Dear Authors,

The manuscript shows interesting  results from a research objectives were the a 3D analysis of the scanned olives. This study may be interesting agricultural fields. I think this article can be of high interest for many scientists. And I have some suggestions to Authors  attached in agronomy-1974446-peer-review-v1 pdf which attached in attachment. also the main suggestion to Authors as described below:

-I suggest to Authors, use statistical analysis to analysis your data,  for validation and comparison of averages. Comparison of the 3D surfaces and 3D solids analysis should be improved, and present your results as figures with  point P values with statistical tests, to relevant items while discussing the results.

-Conclusions should be added in your paper.

- Climates and farming systems, parameters such as  microbial activity can be significant contributors to measured appearance of damage to the fruit.

Round 2

Reviewer 1 Report

Q1. The abstract has not been modified, only some contents have been deleted, and the author has described too much about 2D technology. From the title and aim of the paper, the focus of this paper should fully describe the specific methods, conclusions and advantages of 3D digital analysis technology. It is suggested that the authors continue to supplement and describe the specific methods, conclusions and advantages of 3D digital analysis technology in the abstract.
Q2. In Image Analysis, line 210, "Roundness, to determine the degree of “sharpness” of the corners of the bruised areas". The author states that roundness is related to bruised area, but healthy olives should not have bruised area, so consider whether the definition is correct or not.
Q3. The comparison results of the two technologies are described too much in the section 4 Discussion, which needs to be simplified. 
